# Structure and Function of the T4 Spackle Protein Gp61.3

**DOI:** 10.3390/v12101070

**Published:** 2020-09-24

**Authors:** Shuji Kanamaru, Kazuya Uchida, Mai Nemoto, Alec Fraser, Fumio Arisaka, Petr G. Leiman

**Affiliations:** 1Department of Life Science and Technology, Tokyo Institute of Technology, S2-7 4259 Nagatsuta-cho, Midori-ku, Yokohama, Kanagawa 226-8503, Japan; uchida.k.ab@gmail.com (K.U.); mai.nemo0422@gmail.com (M.N.); fumio.arisaka@gmail.com (F.A.); 2Department of Biochemistry and Molecular Biology, Sealy Center for Structural Biology and Molecular Biophysics, University of Texas Medical Branch at Galveston, TX 77555-0647, USA; alefrase@utmb.edu; 3École Polytechnique Fédérale de Lausanne, Institute of Physics of Biological Systems, 1015 Lausanne, Switzerland

**Keywords:** bacteriophage, lysis, lysozyme inhibitor

## Abstract

The bacteriophage T4 genome contains two genes that code for proteins with lysozyme activity—*e* and *5*. Gene *e* encodes the well-known T4 lysozyme (commonly called T4L) that functions to break the peptidoglycan layer late in the infection cycle, which is required for liberating newly assembled phage progeny. Gene product *5* (gp5) is the tail-associated lysozyme, a component of the phage particle. It forms a spike at the tip of the tail tube and functions to pierce the outer membrane of the *Escherichia coli* host cell after the phage has attached to the cell surface. Gp5 contains a T4L-like lysozyme domain that locally digests the peptidoglycan layer upon infection. The T4 Spackle protein (encoded by gene *61.3*) has been thought to play a role in the inhibition of gp5 lysozyme activity and, as a consequence, in making cells infected by bacteriophage T4 resistant to later infection by T4 and closely related phages. Here we show that (1) gp61.3 is secreted into the periplasm where its N-terminal periplasm-targeting peptide is cleaved off; (2) gp61.3 forms a 1:1 complex with the lysozyme domain of gp5 (gp5Lys); (3) gp61.3 selectively inhibits the activity of gp5, but not that of T4L; (4) overexpression of gp5 causes cell lysis. We also report a crystal structure of the gp61.3-gp5Lys complex that demonstrates that unlike other known lysozyme inhibitors, gp61.3 does not interact with the active site cleft. Instead, it forms a “wall” that blocks access of an extended polysaccharide substrate to the cleft and, possibly, locks the enzyme in an “open-jaw”-like conformation making catalysis impossible.

## 1. Introduction

For most phages, release of progeny virions requires lysis of the host, effected through the controlled activity of at least one phage-encoded muralytic enzyme, the endolysin. Gene *e* of bacteriophage T4 encodes a canonical endolysin-gene product *e* or gp e, which is also called T4L in the literature and here [1]. At the end of the infection cycle of the T4*e* mutant (a T4 mutant defective in the function of gene *e*), virions are trapped in the cytoplasm of the dead host cell [2].

In 1968, Emrich isolated pseudo revertants of T4*e* that carried extragenic suppressor mutations in a gene she called *s*, *sp*, and Spackle in various contexts [3]. The name was derived from the hypothesis that the mutation inactivated a phage-encoded enzyme that participates in the synthesis (s) or repair (‘spackling”, sp) of the peptidoglycan. However, Kao and McClain [4] isolated another class of extragenic suppressors that mapped to the essential tail gene *5* and reported that gp5 had lysozyme activity. This led them to propose that instead of repairing the peptidoglycan layer, Spackle actually protects it from the gp5 lytic activity [5]. The lysozyme activity of gp5 was confirmed later by Nakagawa et al. [6] when they purified gp5 from phage tails. Later, Takeda et al. [7] showed that the temperature-sensitive gene *5* mutant T4-*5^ts1^* originally isolated by Kao and McClain [4] and which suppressed the gene *e* mutation had a single missense mutation G322D.

The function of the Spackle and gp5 proteins in the life cycle of T4 grown in high density cell cultures has been most recently examined by Abedon [8,9,10] with the help of T4-*sp* and T4-*5^ts1^* mutants. The phenotype of the T4-*sp* mutant is identical to that of the wild type (WT) T4 when the cell density and phage titer are such that infection results in a single round of phage multiplication. However, Spackle delays lysis and increases the burst size if the cell is infected by several phages far apart in time. Integrating all observations, Abedon concluded that Spackle’s primary function is to protect the cell from damage to the cell envelope caused by gp5 that originates from virions of phages that infect after the initial infection is already under way. Abedon also suggested that Spackle inhibits the activity of gp5 by direct interaction [8,11].

In 1999, we showed that full-length gp5 undergoes maturational cleavage between Ser351 and Ala352 [12], but both the N- and C-terminal parts (which we later called gp5* and gp5C) are found in the phage particle [12] (Figure 1). In 2002, we determined the crystal structure of gp5 that showed that gp5 consists of three domains: an N-terminal OB-fold (Oligonucleotide/Oligosaccharide-Binding fold, gp5OB-fold, residues 1–129), a lysozyme domain (gp5Lys, residues 174–339), and a needle-like C-terminal β-helical domain (gp5β, residues 389–575) [13]. These domains are connected by long linkers (residues 130–173 and 340–388) but nevertheless form a compact, trimeric, spike-shaped structure. We also found that: (1) gp5OB-fold interacts with the baseplate hub protein gp27 located at the tip of the tail tube and thus is required for incorporation of the protein into the T4 particle; (2) the substrate-binding cleft of gp5Lys is identical to that of T4L; (3) gp5β forms a needle that could pierce the outer membrane of the cell during infection; (4) the maturational cleavage is in the linker connecting gp5Lys with gp5β (gp5*, therefore, consists of gp5OB-fold and gp5Lys) (Figure 1). The G322D suppressor mutation, which abrogates the activity of the Spackle protein, is located in the lysozyme domain at its interface with the needle, but its exact role in the phenotype could not be inferred from the available data.

The identity of the *sp* gene was finally established in 1999 and assigned to gene *61.3* [14]. It encodes a polypeptide of 97 amino acids with an N-terminal secretory signal sequence of 22 residues. We hypothesized that the mature form of Spackle/gp61.3 (residues 23–97) functions in the periplasm to selectively inhibit the activity of gp5, possibly by direct interaction. Accordingly, this interaction must be abolished in the *5^ts1^* (G322D) mutant.

Here, we describe the structure and function of gp61.3 and show that gp61.3 inhibits the lysozyme activity of gp5 but not that of T4L. We show that gp61.3 forms a complex with gp5Lys in such a way that it blocks access of the extended polysaccharide moiety of the peptidoglycan to the catalytic cleft. Furthermore, gp61.3 interacts with gp5Lys while the latter is an “open jaw”-like conformation that is incompatible with catalysis. This mode of regulation constitutes a novel mechanism for lysozyme inhibitors.

## 2. Materials and Methods

### 2.1. Construction of Expression Vectors

The plasmid pUC613, a gift from Yonesaki, T., Osaka University Japan [14], which contained gene *61.3* in the antisense direction under the control of the pUC18-like lac promoter, was digested by *Eco*RI and *Hind*III. The fragment containing gene 61.3 was cloned using the same restriction sites into pUC19 in the sense direction resulting in a plasmid that was named pMN1-1.

Full-length gp5 (Figure 1) was expressed using the plasmid pSZ146 [12], a pET-29a-derivative vector (Merck, Darmstadt, Germany).

pSZ161-2, a plasmid for expression of the gp5^1-372^ fragment (Figure 1) that carried a C-terminal His-tag, was constructed as follows. The pSZ146 vector was first digested by *Xho*I, then treated with the Klenow fragment of *E. coli* DNA polymerase I to repair the overhanging 3′ ends, and subsequently ligated by T4 DNA ligase.

A G332D mutant of gp5 was obtained by site-directed mutagenesis with the help of the inverse PCR technique using the pSZ161-2 vector and the KOD-Plus-Mutagenesis Kit (Toyobo, Osaka, Japan). The PCR primers were 5′-ACCGTGCATCCCGTGTTACCA-3′ and 5′-CTTTTGTTTGTTGATACCAC-3′ (the underlined letters indicate the glycine-to-aspartate codon replacement). The mutation in the resulting vector, pNS2-1, was confirmed by sequencing on a 3730xl DNA Analyzer (Applied Biosystems) at the TIT Bio-Technical Center.

The plasmid for expression of T4L, named pET-gE, was created as follows. Gene *e* of phage T4 was amplified by PCR from the T4 genome with primers that were designed to add a *Kpn*I and a *Bam*HI site at the 5′ and 3-end of the gene. The primers were 5′-GGTGGTACCACTTAGGAGGTATTATGAATATATTT-3′ (the *Kpn*I site is underlined) and 5′-CCCGGATCCCAGCTTTATAGATTTTTATACGCGTCCC-3′ (the *Bam*HI site is underlined). PCR was performed with KOD-Plus-DNA polymerase (Toyobo). The PCR products were purified, digested with *Kpn*I and *Bam*HI and ligated to the vector pET29a, which was cut open with the same enzymes. The insert was verified to have the correct sequence by DNA sequencing.

### 2.2. Protein Expression

For protein production, each of the newly created vectors described above was introduced by chemical transformation into *E. coli* BL21 (DE3). These cells were then grown at 37 °C with rigorous aeration by rotary shaking in the LB (Miller) medium (1% (*w*/*v*) Bacto Tryptone, 0.5% (*w*/*v*) Bacto yeast extract, 1% (*w*/*v*) NaCl). When cell optical density at a wavelength of 600 nm (OD_600_) reached a value of 0.4, expression was induced by an addition of isopropyl-thio-galactoside (IPTG) to a final concentration of 1 mM. Cells were harvested by centrifugation (5000× *g* for 10 min, R9AF rotor, Eppendorf Himac Technologies Co., Ltd., Hitachinaka, Japan) three hours after induction. Cell pellets were frozen in liquid nitrogen and stored at −80 °C. Selenomethinone-labeled proteins were produced by following a standard procedure [15] that involves a methionine auxotrophic strain *E. coli* B834 (DE3) cells and SelenoMet™ Medium (Molecular Dimensions Limited).

### 2.3. Purification of Gp61.3

For this and all other proteins, all purification steps were performed at 4 °C.

The frozen pellet (6 g) of cells that expressed the pMN1-1 plasmid was resuspended in 50 mL of buffer A (50 mM Tris pH 8.0, 150 mM NaCl) on ice. After stirring for 30 min, the cells were spun down at 10,000× *g* for 30 min (R18A rotor, Eppendorf Himac Technologies Co., Ltd.). The supernatant, which mostly contained small periplasmic proteins, was dialyzed in buffer B (20 mM Tris pH8.0) to remove excess salt. The dialyzed sample was applied to a HiTrap Q HP (5 mL, GE healthcare) anion exchange column pre-equilibrated with buffer B. The bound material was eluted with a 0 to 1 M NaCl linear gradient. Fractions that contained gp61.3 were pooled together and applied to a HiLoad 16/600 Superdex 75 pg (GE healthcare) size exclusion column, which was pre-equilibrated with buffer C (20 mM Tris pH 8.0, 100 mM NaCl).

### 2.4. Purification of Gp5*

Earlier, we showed that gp5* and gp5C can be separated by incubation at 37 °C for 2 h or longer (Figure 1) [12]. Hence, to purify gp5*, we extended the previously published protocol for the purification of full-length gp5 [12] with additional steps as described below.

Pellets of cells that expressed the pSZ146 plasmid in 1 L of medium was resuspended in 30 mL of buffer D (100 mM Tris at pH 8.0, 25 mM imidazole) containing 0.2 mM phenylmethylsulfonylfluoride (PMSF) on ice. The cells were then lysed by sonication (Branson Sonifier 250). The lysate was then centrifuged at 28,000× *g* for 30 min (the R18A rotor). The supernatant was loaded onto a 5 mL HisTrap HP (GE healthcare) Ni-affinity column pre-equilibrated with buffer D. The his-tagged full length gp5 protein was eluted from the column with a 25–500 mM imidazole gradient. Eluted fractions were pooled and incubated at 37 °C for 2 h and centrifuged at 28,000× *g* for 30 min (the R18A rotor). The supernatant was loaded on a pre-equilibrated 5 mL HisTrap HP column at 4 °C. gp5* was collected in the flow through fractions.

### 2.5. Purification of Gp5^1-372^ and Gp5_G322D_^1-372^

The pellet of cells that expressed either the pSZ161-2 or pNS2-1 plasmid in 1 L of LB medium was resuspended in 30 mL of buffer D with 0.2 mM PMSF on ice. The cells were then lysed by sonication (Branson Sonifier 250). The lysate was then centrifuged at 28,000× *g* for 30 min (the R18A rotor). The supernatant was loaded onto a pre-equilibrated 5 mL HisTrap HP Ni-affinity column with buffer D. The histidine-tagged protein was eluted from the column with a 25–500 mM imidazole gradient. EDTA was added to each collected fraction to a final concentration of 5 mM to avoid aggregation. Eluted fractions were pooled and applied onto a HiTrap Q HP (5 mL) anion exchange column pre-equilibrated with buffer B. The bound material was eluted with a 0 to 1 M NaCl linear gradient. Fractions containing gp5^1-372^ were further purified with a HiLoad 16/600 Superdex 200 pg (GE healthcare) size-exclusion column using buffer C.

### 2.6. Purification of T4L

The pellet of cells (3 g) that expressed the pET-gE plasmid was resuspended in 25 mL of buffer E (0.1 M sodium phosphate buffer, pH 6.6, 0.2 M NaCl, 10 mM MgCl_2_ and 1 mM CaCl_2_). Then, 0.5 mL of chloroform was added to this mixture and the suspension was stirred for 30 min. To reduce the viscosity, 1 mL of 1 M MgCl_2_ and a few grains of DNase I were added, and the mixture was stirred for an additional 1.5 h. The lysates were centrifuged for 30 min at 28,000× *g* (the R18A rotor) to remove cell debris. The supernatants were dialyzed into buffer F (50 mM Tris pH7.25, 1 mM EDTA). The dialyzed sample was applied to the HiTrap CM FF (5 mL, GE healthcare) cation exchange column, pre-equilibrated with buffer F. The bound material was eluted by 50–300 mM NaCl linear gradient with the same buffer. Fractions containing T4L were pooled, dialyzed against 50 mM sodium phosphate buffer (pH 5.8), and then run through a HiTrap SP HP (1 mL, GE healthcare) cation exchange column equilibrated with the same buffer. T4L was eluted with buffer G (0.1 M sodium phosphate, pH 6.6, 0.55 M NaCl).

### 2.7. Purification of Gp61.3-Gp5Lys Complex

To obtain the gp61.3-gp5Lys complex, a mixture of separately purified gp61.3 and gp5^1-372^ in a 1.2: 1 molar ratio was first subjected to size exclusion chromatography in buffer C on a HiLoad 16/600 Superdex 200 pg column. Fractions containing both proteins were pooled and treated with trypsin in a 100: 1 weight-to-weight ratio for 2 h at 20 °C. After the digestion, size exclusion chromatography was repeated using the same column and buffer, and fractions containing both proteins were pooled.

### 2.8. Crystallization and Data Collection

Purified SeMet gp61.3 was dialyzed in 10 mM Tris buffer, pH8.0, and then concentrated to about 20 mg/mL using a Vivaspin 6 centrifugal filter device (5 kDa cutoff, GE Healthcare). Crystals were obtained in hanging drops at 20 °C using 34% PEG4000, 0.1 M Glycine, pH9.4, 5–10 mM CaCl_2_ as mother liquor. Crystals were soaked in the mother liquor containing 25% ethylene glycol as a cryo-protectant and were then cooled in a nitrogen stream at 100 K. Diffraction data was collected at 100 K using an ADSC Q315 CCD detector on the ESRF beam line BM30. The wavelength of the X-ray beam was chosen to maximize the anomalous scattering of selenium atoms at the Se K-edge. Crystals diffracted to ~1.6 Å resolution (Table 1).

Prior to crystallization, the gp61.3–gp5Lys complex was dialyzed in 10 mM Tris buffer, pH8.0, and then concentrated to about 20 mg/mL using a Vivaspin 6 centrifugal filter device (50 kDa cutoff). Crystals used for X-ray diffraction studies were grown by vapor diffusion in a hanging drop at 20 °C using 33% PEG550MME, 150 mM MES, pH6.4, 50 mM KSCN as mother liquor. Diffraction data was collected at 100 K using a Mar225 detector on the SLS beam line PXIII. Crystals diffracted to ~1.15 Å resolution (Table 1).

### 2.9. Structure Determination

For SeMet gp61.3, all data sets were indexed, integrated, and scaled using the program HKL2000 [17]. The positions of selenium atom and the initial phases were determined with the program SHELXD [16]. These phases were improved by density modification using the program Parrot [18]. Subsequently, most of the atomic model was built automatically by the ARP/wARP program [19]. Missing parts and the solvent structures were built manually using Coot [20]. The model was refined with the help of Refmac5 [21], Phenix [22], and Coot programs (Table 1).

For the gp61.3–gp5Lys complex, all data sets were indexed and integrated by Mosflm [23] and scaled using the program SCALA [24]. Crystallographic phases were determined by the molecular replacement with the help of the Phaser program [25], using the crystal structures of gp61.3 and gp5Lys [13] as search models. Subsequently, the atomic model was built with Coot. The model was refined by Refmac5, Phenix, and Coot (Table 1).

The atomic structures and structure factors of gp61.3 and the gp61.3–gp5Lys complex were deposited into the Protein Data Bank under the accession numbers 7CN6 and 7CN7, respectively.

### 2.10. Lysozyme Halo Assay

*E. coli* BE cells were mixed with a soft agar, overlaid onto an LB agar plate and incubated at 37 °C for 16 h, resulting in an agar layer impregnated with *E. coli* bacteria. The bacterial lawn was then exposed to chloroform vapors for 30min at 25 °C. For this, chloroform was added to the lid of the plate to form a continuous layer, and the cells were incubated above it. The chloroform was then removed, and 5 µL aliquots of protein solution or buffer were spotted on the chloroform-treated *E. coli* lawn. In mixtures containing gp61.3 (gp5:gp61.3 and T4L:gp61.3), it was present in a 1.2 molar excess. The plates were incubated at 37 °C until the control lysozyme spot became clear.

### 2.11. Analytical Ultracentrifugation

Analytical ultracentrifugation was performed with the help of the Optima XL-I (Beckman-Coulter) ultracentrifuge using an eight-hole An50Ti rotor at 20 °C. The dialysis buffer or loading buffer for size exclusion chromatography was used as a reference solution. Sedimentation velocity data was collected with the centrifugal force at 181,714× *g*. Moving boundaries were recorded at a wavelength of 280 nm. The sedimentation coefficient distribution function, c(s), was obtained with the SEDFIT program [26]. The distribution of molecular weight, c(M), was obtained by converting c(s) to c(M) as implemented in the SEDFIT program.

### 2.12. Cell Lysis Assay

The lytic activity of gp5 and T4L associated with the expression of these proteins in the cell was tested as follows. Single colonies of *E. coli* BL21 (DE3) harboring the vectors pSZ202-2 (gp5), pETgE (T4L) or pET29a (control) were grown in LB medium at 37 °C with aeration by orbital shaking at 200 rpm. Protein expression was induced by the addition of IPTG to a final concentration of 1 mM when the turbidity of the cell culture, as measured by OD at 600 nm, reached a value of ~0.5. After induction, the cultures continued to be cultivated with the same parameters and their turbidities were monitored by OD_600_. To test whether gp5 is capable of lysing cells from the outside, purified gp5 was added (to a final concentration of 0.1 mg/mL) to a culture harboring the pET29a vector when the OD_600_ of the culture was about 0.5.

### 2.13. Molecular Dynamics

The crystal structure of the gp61.3-gp5Lys complex was solvated and ionized in a water box using VMD [27]. Following this, NAMD2 [28] conjugate gradient energy minimization was applied to the system while incrementally removing constraints for a total of 200,000 steps. The system was then heated to 300 K in 5 K increments using a Langevin thermostat [29] for a total of 60 ns. In a final pre-equilibrium step, the system was equilibrated for 20 ns under constant pressure and temperature (the “NPT” ensemble) using Nosé-Hoover Langevin pressure control [30,31]. Then, constant velocity-steered molecular dynamics (SMD) [32] was used to pull gp61.3 away from gp5Lys at a speed of 1 Å/ns along the line connecting the center of masses (COMs) of the two molecules. The pull force was applied to all gp61.3 Cα atoms with a spring constant of 50 kcal/mol/Å^2^. The structure of gp5Lys was constrained using harmonic restraints on all Cα atoms. The SMD protocol was executed in the NPT ensemble for 20 ns. The distance between COMs of gp5Lys and gp61.3 changed from 19.8 Å (the initial equilibrated complex) to 38.6 Å. However, after 24.8 Å of separation, the SMD forces decreased dramatically and likely corresponded to the drag force of pulling gp61.3 through solvent. For this reason (and to limit computational effort), the (19.8 Å, 24.8 Å) COM separation interval was chosen for further adaptive biasing force (ABF) analysis. Ten SMD simulation snapshots in which the distance between gp61.3 and gp5Lys COMs spanned the (19.8 Å, 24.8 Å) interval in steps of 0.5 Å were chosen as starting points for the adaptive biasing force (ABF) calculations [33,34]. The ABF calculations were run for ~250 ns in the NPT ensemble. All simulations were performed using NAMD2 and used the CHARMM36 forcefield [35].

### 2.14. Sequence Alignments and Molecular Graphics Generation

Figures 6a,b, 7b,c,d, 8b,c,e,f, 9b–h and 11a,b were rendered by UCSF Chimera [36]. Figures 5, 7a,b and 8a,d were rendered by MolFeat (FiatLux, Tokyo, Japan). In Figure 9a, the T4L and gp5 sequence were aligned by Clustal W [37] and visualized by the Snapgene viewer (Insightful Science; available at snapgene.com).

## 3. Results

### 3.1. Gp61.3 is Translocated into the Periplasm and Its Signal Sequence Is Cleaved

Infection of an *E. coli* cell by T4 and T4-induced lysis of the cell at the end of the infection cycle are complex processes that involve many players [8,38]. The functions of some participants have been derived from indirect assays, such as, for example, the morphology of lysis plaques, which is determined by a myriad of factors. The gp61.3 Spackle protein was proposed to inhibit the gp5 lysozyme activity [5] but its exact function and cellular localization remained unknown.

We overexpressed gp61.3 in *E. coli* (Appendix A), pelleted the cells by centrifugation, froze the pellet at −80 °C, and thawed it. This procedure did not lyse the cells but instead released small-to-medium-sized proteins from the periplasm [39,40] (Figure 2, Lanes 1 and 2). Gp61.3 was a major component in this mixture (Figure 2, Lane 2). Gp61.3 could then be purified to homogeneity by successive round of column chromatography (Figure 2, Lane 3). N-terminal peptide sequencing of purified gp61.3 showed that in agreement with the hypothesis proposed by Kai et al., [14] the first 22 residues of gp61.3 comprised a signal peptide that was cleaved in the protein released from the periplasm.

### 3.2. Gp61.3 Forms a 1:1 Complex with Gp5*

Previous work suggested that gp61.3 interacts with gp5 [8,11]. We used analytical ultracentrifugation (AUC) to examine the nature of this interaction and to establish the composition of the gp61.3-gp5 complex.

AUC analysis of gp5*:gp61.3 mixtures at different molar ratios is presented in Figure 3. On their own, gp5* and gp61.3 had sedimentation coefficients of 2.99 S and 1.38 S, respectively. When both proteins were present in the mixture, a peak corresponding to a sedimentation coefficient of 3.6 S appeared. This peak was partially obscured by the gp5* peak if gp5* was in excess in the mixture.

The sedimentation coefficient of 3.6 S corresponds to a molecular weight (MW) of 45 kDa. This value is close to the sum of the MWs of one gp5* and one gp61.3 molecule (47.5 kDa). Thus, gp61.3 and gp5* form a 1:1 complex in solution. A similar analysis was performed for a T4L:gp61.3 mixture but no complex was detected. This finding was additionally confirmed with size exclusion chromatography (Appendix A).

To further delineate the interaction between gp5 and gp61.3, while taking into account that in addition to the lysozyme domain, the interdomain linkers of gp5 can participate in the interaction with gp61.3, we cloned and purified a fragment of gp5 comprising residues 1–372 (gp5^1-372^) that is slightly larger than gp5* (Figure 2, Lane 4). Gp5^1-372^ (Figure 2, Lane 4) and gp61.3 (Figure 2, Lane 3) were mixed, the mixture was purified and treated with trypsin. We surmised that it should be possible to find proteolysis conditions in which all parts of gp5^1-372^ that interact with gp61.3 would be protected while non-interacting parts would be removed, resulting in a compact complex suitable for crystallization.

A proteolysis protocol, in which gp61.3 appeared to be unaffected by trypsin while gp5^1-372^ was digested, has been established (Figure 2, Lane 5). The N-terminal hexapeptide of trypsin-digested gp5^1-372^ was found to be PLSEIP, which matched the sequence of gp5 starting from residue Pro162. The C-terminal cleavage site has not been established. The last gp5 residue with an interpretable electron density in the crystal structure of the gp61.3-gp5Lys complex (see Section 3.5) is Glu342. Hence, besides Ser372, either Lys344, Arg348, or Lys359 could form the C-terminal residue in trypsin-treated gp5^1-372^. Despite this uncertainty and disregarding the few extra residues of the interdomain linkers, we will call this gp5^162-372^ fragment gp5Lys everywhere in the text (Figure 1).

### 3.3. Gp61.3 Inhibits Lysozyme Activity of Gp5 but Does Not Affect the Activity of T4L In Vitro

To examine whether gp61.3 can inhibit gp5 lysozyme activity, we used a spot test assay on live *E. coli* cells that were treated with chloroform vapors. Chloroform creates large holes in the outer membrane of the cell. This allows the lysozyme to reach the peptidoglycan layer and lyse the cell.

gp5^1-372^, T4L, and the gp5_G322D_^1-372^ mutant that carried the G322D mutation found in the *5^ts1^* suppressor allele, all created a lysis spot in the *E. coli* lawn (Figure 4). When the three proteins were mixed with gp61.3 in a 1:1.2 ratio (gp61.3 in 20% excess), the lysis activity of T4L and that of the gp5_G322D_^1-372^ mutant appeared to be unaffected, whereas the lysis spot of gp5^1-372^ was barely detectable (Figure 4). Taking into account the complex formation results presented above, we conclude that gp61.3 inhibits the activity of its lysozyme partner by direct interaction. No such interaction is formed with either T4L (Appendix A) or, by extension, with gp5_G322D_^1-372^. The latter finding explains the phenotype of the T4-*5^ts1^* phage mutant that carries inactive T4L but lyses cells normally, similar to the WT T4 [8].

### 3.4. The Crystal Structure of Gp61.3 Shows That It Has a Novel Fold

The structure of gp61.3 was determined by a single wavelength anomalous diffraction technique using a Se-methionine substituted protein. To maximize anomalous scattering of Se atoms, an X-ray wavelength near the Se K-edge absorption line was used (Table 1). A 1.6 Å resolution crystal structure shows that gp61.3 is a fully α-helical globular protein (Figure 5). It contains 5 α-helices: helix 1 (26–34), helix 2 (37–55), helix 3 (57–70), helix 4 (73–75) and helix 5 (87–95) that are connected by short loop linkers.

There are three gp61.3 monomers in the crystallographic asymmetric unit. Barring different rotamers, their structures are nearly identical. The root mean square deviations (RMSDs) found on pairwise superpositions, which involved all 75 Cα atoms of gp61.3 are 0.24 Å, 0.25 Å and 0.35 Å.

In two of the three gp61.3 molecules contained in the crystallographic asymmetric unit, a calcium ion is coordinated by several water molecules and the main chain hydroxyl groups of residues Val94 and Glu97. The identity of the ion was derived from the site’s geometry (the structure of the coordination sphere and bond lengths) and the height of the electron density peak. Moreover, calcium was present in the crystallization solution (see Materials and Methods).

The thiol groups of two cysteine residues Cys29 and Cys81 are close in space, but not close enough to form a proper disulfide bond (Figure 6a). The distance between the sulfur peaks in the electron density is 2.43 ± 0.06Å when averaged over the three molecules contained in the asymmetric unit. As a global average of billions of molecules comprising the crystal, the crystallographic electron density most likely depicts a superposition of the conformation with a disulfide bond (which has a length of 2.05 Å) and a vast number of conformations in which the thiol groups are greater than the Van der Waals distance away from each other. The non-bonded conformations comprise a significant fraction of this ensemble as the average S–S atom distance is much greater than the dictionary value for disulfide bond.

A search for proteins with gp61.3-like folds using the DALI server [41] results in a large number of hits with Z-scores greater than 2.0. However, none of these proteins contains a gp61.3-like domain, which we define as a fragment that possesses its own hydrophobic core. The best DALI hit with a Z-score of 5.6 is the SecA ATPase from *Thermus thermophilus* (PDB code 2IPC), a cytoplasmic protein that is responsible for posttranslational translocation of polypeptide substrates through the SecY channel in the cytoplasmic membrane. The superposition involves nearly the entire structure of gp61.3 with 70 equivalent Cα atoms in the alignment and results in an RMSD of 2.8 Å and a sequence identity of 9%. However, the gp61.3-like part of SecA belongs to two domains—an α-helical Scaffold domain (gp61.3 helices 1, 4, 5) and an α-helical Wing domain (gp61.3 helices 2 and 3) [42]. Furthermore, the lengths of SecA helices are more than double those of gp61.3. Thus, on the one hand, the similarity to a protein involved in the translocation of proteins across the plasma membrane appears to be nonincidental, but on the other hand, the matching fragment of the SecA ATPase is not a complete domain and the superposition sequence identity is extremely low. Hence, even though gp61.3 is a very small protein, it appears to possess a novel and possibly unique fold.

### 3.5. The Crystal Structure of the Gp61.3–Gp5Lys Complex

To understand the interaction between gp61.3 and gp5Lys, we crystallized the complex of gp61.3 and gp5Lys obtained as described above (Figure 2, Lane 5), solved its crystal structure by Molecular Replacement and refined it to a resolution of 1.15 Å (Table 1). The complex consists of one molecule of gp61.3 and one molecule of gp5Lys. The crystal structure contains a fragment of a poly (ethylene glycol) methyl ether molecule (a chemical that was used in the crystallization), four ethylene glycol molecules, three sodium ions, one chloride ion and 382 water molecules. The ligands were identified based on the appearance of their electron density and chemical environment. The sedimentation coefficient calculated by HYDROPRO [43] using the crystal structure was 3.66 S, which agrees well with the value determined by AUC (Figure 3).

The interface between gp5 and gp61.3 contains 10 hydrogen bonds and 8 salt bridges (Figure 7a). It involves 16 water molecules and no ions. The buried surface area is 1733.1 Å^2^. Surprisingly, PISA [44] categorizes the gp61.3-gp5 complex as metastable. The gp61.3-gp5 interface has a favorable free energy of −3.3 kcal/mol, but the free energy of dissociation is also favorable at −0.4 kcal/mol. The gp61.3-gp5 complex survives the environment of *E. coli* cells treated with chloroform (Figure 4) and as such, the interface was expected to be more favorable.

Considering that PISA analysis contradicts the experimental observations, we quantified the free energy of association for the gp61.3-gp5Lys complex with the help of adaptive biasing force (ABF) molecular dynamics simulations [33,34]. This approach consists of breaking the complex apart using steered molecular dynamics, dividing this trajectory into regions around intermediate structures (named windows), and then calculating the forces (ABFs) necessary to hold the system within each window. Over a sufficiently long simulation, the resulting ABF forces, which compensate for the latent energetics of the system, can be integrated to yield the underlying Hamiltonian [45].

The resulting potential of mean force (PMF) (Figure 7b) for the association of gp61.3-gp5Lys shows a favorable free energy of about −14 kcal/mol. This value is consistent with a complex that is stable enough to withstand the periplasmic milieu and is similar to the association energies measured experimentally for other periplasmic proteins of a similar size [46].

The conformations of both gp5Lys and gp61.3 differ slightly, but significantly, relative to their pre-complex states. Free and gp5-bound gp61.3 molecules superimpose with an RMSD of 0.82 Å between all Cα atoms. Residues Gly96 and Glu97 display the largest deviations of 0.92 Å and 1.38 Å, respectively. Both these residues are part of the gp61.3-gp5 interface. Glu97 of gp61.3 does not bind a calcium or any other ion in the complex. Accordingly, calcium ions did not affect gp61.3-gp5 complex formation (Appendix A). The sulfur atoms of the two cysteines (Cys29 and Cys81), that were too far apart to form a disulfide bond in free gp61.3, are now only 2.16 Å apart (Figure 6b). This is close to, but still slightly longer than, the ideal value of a disulfide bond (2.05 Å). This bond does not appear to play role in gp61.3-gp5 complex formation in vitro as the complex was formed in the presence of DTT (Appendix A). Notably, the crystal structure of gp61.3 carrying a C-terminal His-tag and crystallized in different conditions has just been reported elsewhere [47]. In that case, residues Cys29 and Cys81 have been described as being linked by a disulfide bond (albeit of an unspecified length) and no ions have been mentioned to be located near Gly96 or Glu97. These observations further support the supposition that neither calcium nor the Cys29–Cys81 disulfide bond plays a role in gp61.3-gp5Lys complex formation.

The gp5 lysozyme domain in the gp5 full-length trimer and that in the gp61.3-gp5 complex can be superimposed with an RMSD of 1.14 Å between 175 equivalent Cα atoms. The Cα deviations, however, are distributed along the polypeptide chain in a non-random manner. The two jaw-like subdomains of gp5Lys (Figure 7a) are further away from each in the gp61.3-gp5 complex compared to that in the gp5 trimer. The corresponding distances between the center of masses (COMs) of these subdomains in the two conformations are 24.6 Å and 25.4 Å for the free and gp61.3-bound states of gp5Lys, respectively (Figure 7c,d). Thus, upon interaction with gp61.3, gp5Lys becomes “frozen” in an open jaw-like state.

### 3.6. The Critical Location of G322D Mutation in Gp5Lys

The crystal structure of the gp61.3-gp5 complex shows how the G322D mutation in gp5Lys interferes with the formation of the complex. Gly322 fits against the C-terminus of helix 5 of gp61.3 (which forms the C-terminus of the protein) so that it leaves no space for a side chain (Figure 8a,b). A hypothetical Cβ atom (an alanine side chain) would be wedged between the main chain carboxyls of Glu93 and Val94 with unfavorable distances of 2.7 Å and 1.9 Å, respectively (Figure 8c). A larger side chain (e.g., like an aspartate of the G322D mutant) creates additional unfavorable interactions and clashes (Figure 8c). For this reason, any residue except for a glycine, will likely interfere with the formation of the gp61.3-gp5Lys complex. Consequently, this is one of the most conserved residues in gp5 orthologs (Figure 9a). The equivalent residue in T4L and its orthologs is Asn (Figure 9a). This residue is likely important for gp61.3 to distinguish gp5Lys from T4L.

The structure of the gp5 trimer suggests an explanation as to why the G322D mutation in gp5Lys does not interfere with the lysozyme activity of gp5_G322D_^1-372^ but allows for the assembly of the full-length gp5_G322D_ trimer. The mutation is located at the gp61.3-gp5 interface, far away from the active site (Figure 7c), so it is unlikely to play a role in substrate binding or cleavage. In the gp5_G322D_ trimer structure, the D322 aspartate side chain is at the gp5Lys-gp5β interface, but there is sufficient space for it to point into solution, thus minimizing the amount of unfavorable interactions (Figure 8d–f)). For this reason, the G322D mutation is likely to have a minimal effect on the folding of the gp5_G322D_ trimer, which nevertheless manifests itself in a temperature-sensitive phenotype of the T4-*5^ts1^* mutant. Gp5 participates (and this is absolutely required) at the earliest steps of T4 tail assembly [48], and the efficiency of gp5 trimer formation, which could be moderately impaired in the T4-*5^ts1^* mutant, likely influences the assembly pathway of the entire particle.

The extent of molecular surfaces participating in the gp61.3-gp5 interface is slightly smaller than the size of conserved patches on the surfaces of gp61.3 and gp5 (Figure 9b–e). There is a protrusion that is formed in part by G322 on the surface of gp5. This protrusion fits into a cavity near Val94 on the surface of gp61.3. The two interacting surfaces display complementary charges with the gp61.3 surface being negatively charged (Figure 9f,g). The G322D mutation neutralizes a strong positive charge on the gp5 surface but the rest of the surface charge distribution is undisturbed (Figure 9h). Hence, the clash of the D322 side chain with the interface residues of gp63.1 and V94 in particular (Figure 7c) likely plays a more important role in abolishing complex formation than charge neutralization.

### 3.7. Gp5 Can Penetrate the E. coli Inner Membrane from the Inside on Its Own

In the original T4*e* mutant, which was isolated by Emrich [3], the function of T4L was most likely performed by gp5. Despite having similar levels of enzymatic activity in vitro [6], the two proteins, however, are not equivalent in terms of their activity in vivo and an *E. coli* cell reacts to their presence differently. Overexpression of T4L does not cause cell lysis (Figure 10, filled triangles). To the contrary, overexpression of gp5 leads to lysis that starts about 90 min after IPTG induction (Figure 10, filled circles). This effect is not seen if gp5 is added to the culture medium (Figure 10, empty circles).

In the course of normal infection, T4 expresses a holin protein (gp t) that functions to create openings in the inner membrane [50]. These “holes” allow T4L, which is unable to cross the cytoplasmic membrane on its own, to reach the peptidoglycan layer. Our experiments show that gp5 is capable of translocating through the inner membrane without a holin (Figure 10, filled circles). The gp5β domain is likely to be responsible for this remarkable property as it can cross the plasma membrane on its own [51].

## 4. Discussion

Previous functional analysis of T4 gene *sp* and *5* mutants [5,9] suggested that the Spackle protein gp61.3 is a gp5-specific inhibitor [14]. All our experimental findings support this assertion. Using purified proteins, we showed that gp61.3 and gp5Lys interact directly (Figure 2, Figure 3 and Figure 7) and the stoichiometry of this complex is 1:1 (Figure 3). We also demonstrated that gp61.3 selectively inhibits WT gp5, but not T4L or the gp5_G322D_ mutant (Figure 4). Finally, our atomic-resolution crystal structure of the gp61.3-gp5Lys complex clearly shows how gp61.3 interacts with gp5Lys and provides an explanation as to why G322 is such a critical residue in gp5 (Figure 7). Any other side chain, even as small as that of an alanine and certainly that of an aspartate found in the gp5_G322D_ mutant, will create unfavorable interactions at the gp61.3-gp5Lys interface (Figure 8c) and will interfere with complex formation.

We also examined whether the presence of gp61.3 in the periplasm influences the infection of WT T4 and found no difference in the phage titer in the presence or absence of overexpressed gp61.3. We also showed that gp5, but not T4L, translocates from the cytoplasm to the periplasm during recombinant expression and causes cell lysis. Combining all our results with previous work on gene *sp* and *5* mutants, we propose that the main function of the Spackle protein gp61.3 is to inhibit “free” copies of gp5 that are not incorporated into the virion during T4 particle assembly and thus can escape into the periplasm.

Previous work showed that gp61.3 plays an important role in the inhibition of lysozyme activity of gp5 proteins delivered into the periplasm during tail tube penetration by multiple phage particles at various moments in time [8]. This role of gp61.3 is supported by the high conservation of the gp61.3-gp5Lys interface (Figure 9a,d,e). Cells infected with T4 and expressing gp61.3 can reduce the infection efficiency of a great number of phages that carry gp5Lys domains to which gp61.3 can bind.

By combining the information about the structure, function and conformational changes of T4L [52] with our new structural data on gp5Lys, we propose that the mechanism of gp5Lys activity inhibition by gp61.3 involves two components: a steric hindrance to polymeric substrate binding and a catalytic cycle lock.

The location of the substrate on the gp5Lys molecule can be derived from that of T4L by superimposing the structure of the T4L mutant with a repeating unit of the peptidoglycan covalently linked to its active site (PDB code 148L, [1]) onto gp5Lys (Figure 11a or Figure 10). Both T4L and gp5Lys bind the sugar moiety of the peptidoglycan substrate inside a long cleft between their jaw-like domains (Figure 7a). Both sides of this cleft are open to solution to accommodate the long polymeric substrate. In the gp61.3-gp5Lys complex, gp61.3 forms a wall at one of the cleft’s exits (Figure 10), which likely abolishes the endoglycosidase activity of gp5Lys or turns it into an exoglycosidase with a much-reduced turnover rate of polymeric substrate (the peptidoglycan). This constitutes the first component of the inhibition mechanism.

To understand the second component of the inhibition mechanism, several conformations of gp5Lys and T4L must be compared. The T4L mutant with the covalently linked fragment of the peptidoglycan (PDB code 148L [1]) can be superimposed onto gp5Lys in the gp5 trimer (PDB code 1K28 [13]) and onto the gp61.3-bound gp5Lys with RMSDs of 1.07 Å and 1.47 Å, respectively, with 96% of all Cα participating in the alignment. Substrate-free WT T4L (PDB code 2LZM, [53]) shows a similar trend and superimposes onto gp5Lys in the gp5 trimer and onto gp61.3-bound gp5Lys with RMSDs of 1.05 Å and 1.18 Å, respectively. Thus, the conformation of gp5Lys in the gp5 trimer is closer to the free and substrate-bound conformations of T4L rather than its conformation in the gp61.3–gp5Lys complex. T4L is known to move its two jaw-like domains during the catalytic cycle [54,55], and gp5Lys, having a nearly identical structure, is also likely to “gnaw” on its substrate by moving its “jaws” in a similar manner. The binding of gp61.3, however, likely puts a break on this motion and locks gp5Lys in an open jaw conformation (Figure 7c,d, Appendix A). Therefore, gp61.3 inhibits gp5Lys allosterically by stabilizing one conformation in the gp5Lys catalytic cycle, thus breaking it.

It is interesting to note that the lysozyme activity of gp5 in its trimeric form is 10-fold lower than that of monomeric gp5* [56]. The inhibition is due to the presence of gp5C, which trimerizes the gp5*-gp5C complex (Figure 1). Gp5C is required for membrane piercing but its role in the penetration of the peptidoglycan remains uncertain. Gp5C interacts with gp5Lys in several places including a surface patch that overlaps with the gp61.3 binding site (Figure 8a,d). Thus, gp61.3 cannot bind to gp5Lys when gp5C is present and gp5 is in its trimeric form. Dissociation of gp5C on the one hand activates gp5Lys, but on the other hand it frees the binding interface for gp61.3, which would immediately inhibit it. This complex interplay of events requires additional studies.

We would like to conclude that allosteric inhibition of gp5Lys by gp61.3 in which no part of gp61.3 directly interacts with the substrate binding cleft appears to constitute a novel mechanism. All previously characterized transglucosylase inhibitors (PliG, Ivy, MliC, Tgi2) [57,58,59,60] interact with the substrate binding pocket directly. The significance of this novel inhibition mechanism for the function of gp61.3 and gp5 remains to be understood.

## Figures and Tables

**Figure 1 viruses-12-01070-f001:**
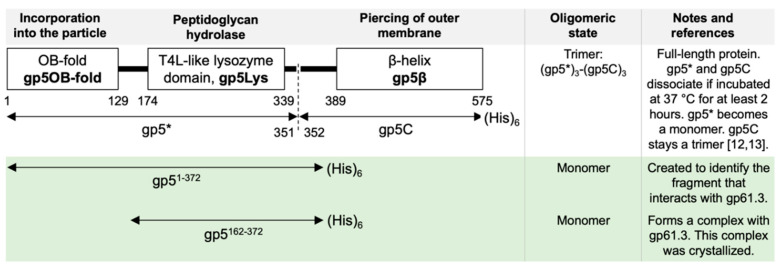
Domain organization of gp5, fragments used in the study, their properties, and additional notes. The diagram of domain organization and experimental notes concerning full-length gp5 are described in detail in [12,13]. Newly created constructs used in this study and the relevant notes are highlighted with a light green background.

**Figure 2 viruses-12-01070-f002:**
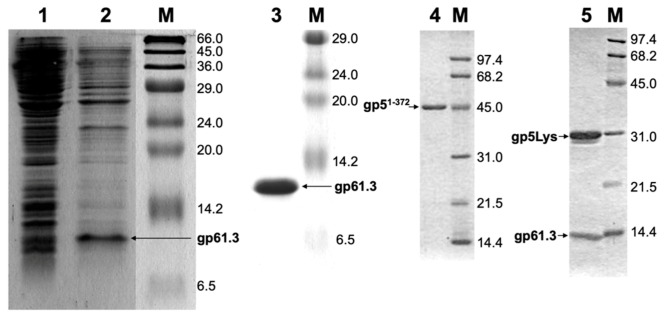
Purification of gp61.3, gp5^1-372^, and the gp61.3-gp5^162-372^ (gp61.3-gp5Lys) complex. All panels show Coomassie Blue-stained Sodium dodecyl sulfate polyacrylamide gel electrophoresis (SDS PAGE). Lane 1: the whole cell lysate of cells expressing gp61.3. Lane 2: the supernatant resulting from a freeze and thaw extraction procedure. Lane 3: purified gp61.3. Lane 4: purified gp5^1-372^. Lane 5: purified gp61.3-gp5^162-372^ (gp61.3-gp5Lys) complex. Note that all fragments of gp5 migrate much slower than their predicted molecular weights. All lanes labeled M are molecular weight standards (see M&M for detail).

**Figure 3 viruses-12-01070-f003:**
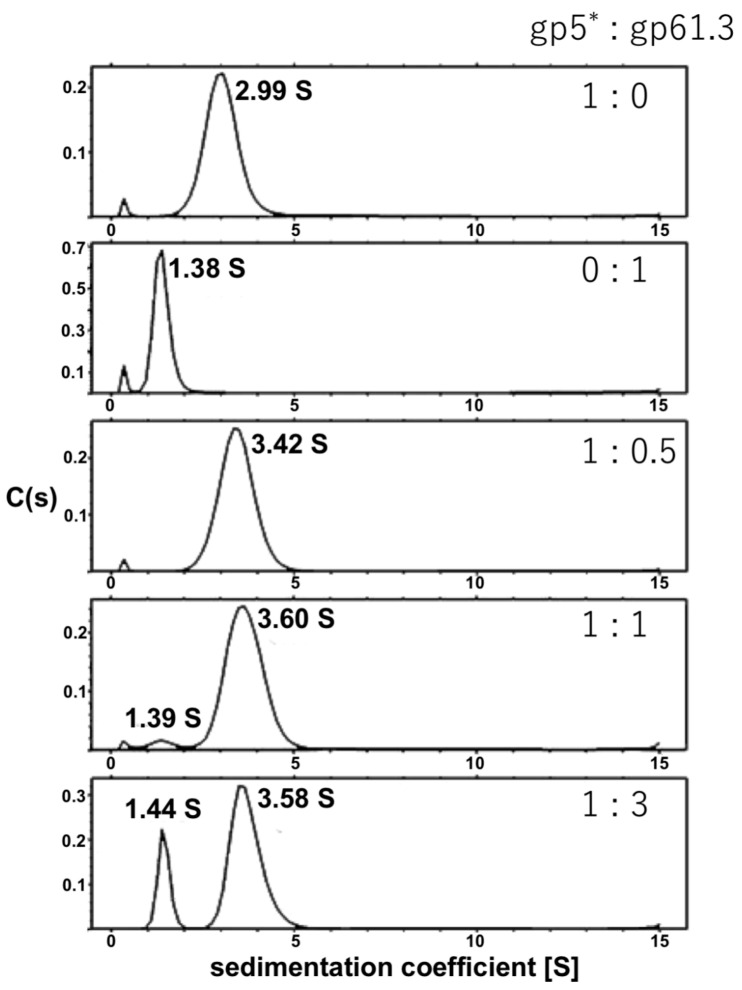
Sedimentation velocity analysis of gp5*:gp61.3 mixtures at different molar ratios (see M&M for detail).

**Figure 4 viruses-12-01070-f004:**
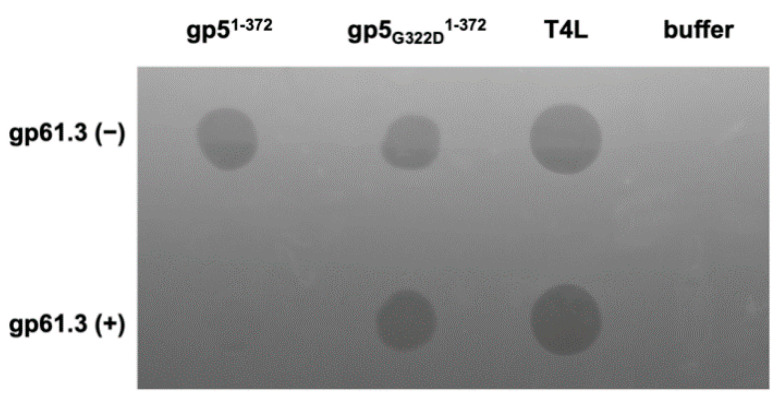
Lysozyme halo formation assay with chloroform vapor-treated *E. coli* lawn. Purified gp5^1-372^, gp5_G322D_^1-372^ and T4L were spotted with (bottom row) or without (top row) gp61.3 (see M&M for detail).

**Figure 5 viruses-12-01070-f005:**
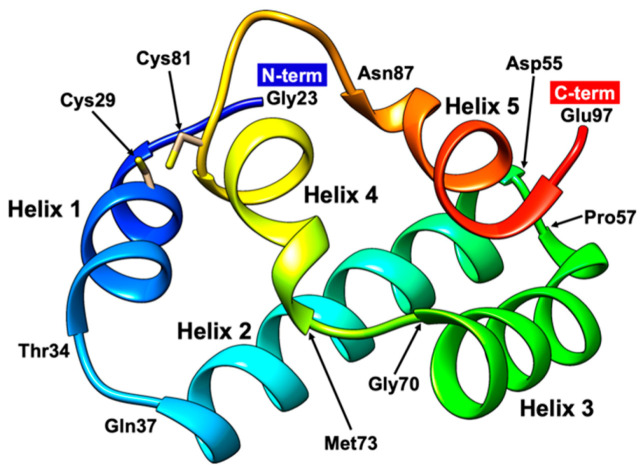
1.60 Å crystal structure of gp61.3. Molecule is colored in rainbow from blue (N-term) to red (C-term). The start and end residues for each helix are indicated.

**Figure 6 viruses-12-01070-f006:**
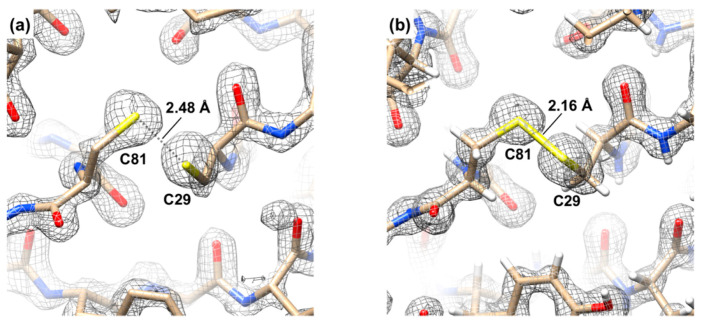
A fragment of the sigma-A weighted 2mFo-DFc electron density of the 1.6 Å resolution gp61.3 SeMet crystal structure (**a**) and that of the 1.15 Å resolution gp61.3-gp5Lys crystal structure (**b**). The map is contoured at 2 standard deviations above the mean.

**Figure 7 viruses-12-01070-f007:**
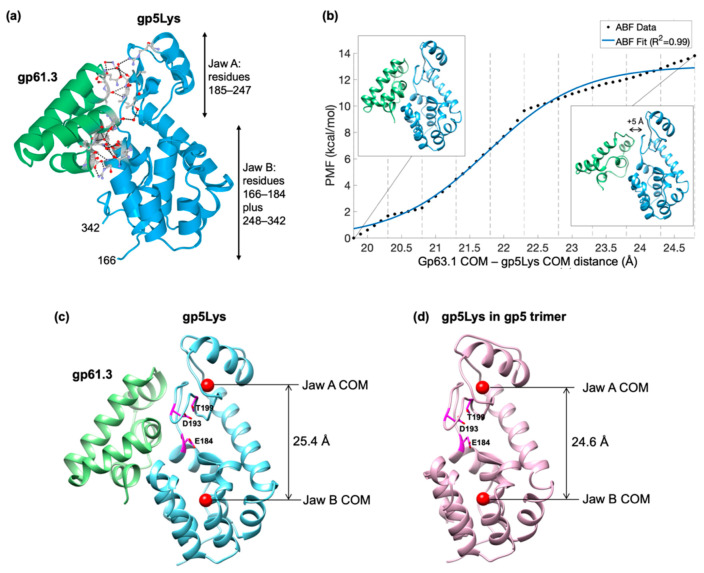
The structure and energetics of the gp61.3-gp5Lys complex. (**a**), a ribbon diagram of the gp61.3-gp5Lys complex. Gp61.3 is colored green and gp5Lys is colored sky blue. Side chains and water molecules forming the interface are shown in the ball-and-stick representation. The two “jaw” domains of gp5Lys are labeled. (**b**) Potential of mean force curve generated from ABF simulations. The simulations covered a distance of COM separations from 19.8 to 24.8 Å, which corresponded to the equilibrium position in the original complex to that plus additional 5 Å. A logistic function, which is the expected shape of the ABF curve, is fit to the raw ABF data. The structure on the left represents the NAMD2-equilibrated gp61.3-gp5Lys complex, whereas the structure on the right represents the dissociated complex (where the COMs are separated by an additional 5 Å relative to the equilibrated state). Dashed horizontal lines correspond to the window boundaries used in the ABF calculations. (**c**,**d**). The distance between the COMs (red spheres) of the two jaw domains in gp5Lys bound to gp61.3 and that in gp5Lys in the gp5 trimer. The active site residues (Glu184, Aps193, Thr199) are shown in the stick representation and colored magenta.

**Figure 8 viruses-12-01070-f008:**
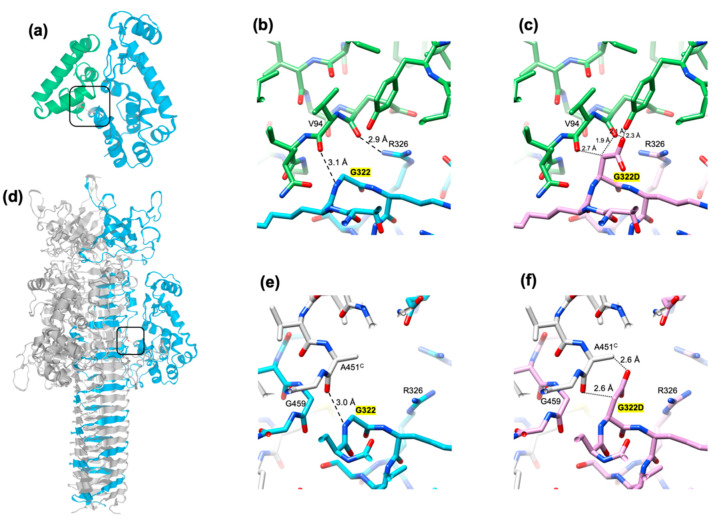
The critical location of the G322D mutation in gp5Lys. (**a**) A ribbon diagram of the gp61.3-gp5Lys complex. The rectangle indicates the location of G322 in the structure of the complex and the area of focus shown in panels (**b**) and (**c**). (**b**) The interaction between gp61.3 and gp5Lys in the vicinity of G322. The Cα atoms of gp61.3 are colored green. (**c**) G322 is mutated to an aspartate and a rotamer with the fewest clashes is shown. (**d**) A ribbon diagram of the gp5 trimer. One chain is colored blue, the other two are gray. The rectangle indicates the area of focus shown in panels (**e**) and (**f**). (**e**) The interaction between gp5Lys and gp5β in the vicinity of G322. (**f**) G322 is mutated to an aspartate. The side chain conformation is one of the most preferred rotamers. For visualization clarity, the orientation of the molecules in panels (**b**,**c**,**e**,**f**), is different to that shown in panels (**a**) and (**b**). In panels (**b**,**c**,**e**,**f**), the dashed and dotted lines indicate hydrogen bonds and unfavorable contacts, respectively.

**Figure 9 viruses-12-01070-f009:**
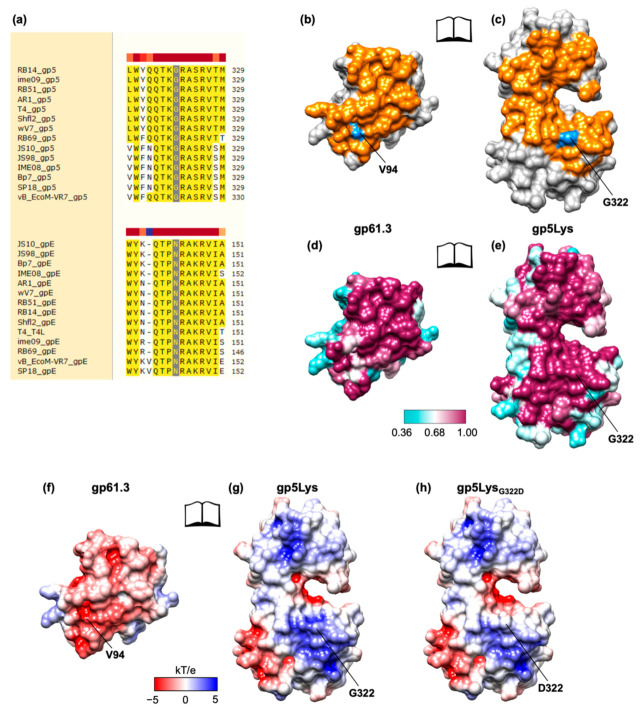
Properties of the gp61.3-gp5Lys interface. (**a**) A fragment of sequence alignment of gp5 proteins from T4-like phages near residue G322 and the same for T4L orthologs (labeled gpE here) around the equivalent residue N144. The G322 and N144 columns are marked with a gray color. (**b**) and (**c**): the molecular surface of the gp61.3-gp5Lys interface in an “open book” representation. The interacting residues are colored orange. G322 of gp5Lys and its most proximal residue of gp61.3 (V94) in the gp61.3-gp5Lys complex are colored dodger blue. (**d**) and (**e**): sequence conservation is mapped on the molecular surface of the gp61.3 and gp5Lys interface (an open book representation). The color bar gives the degree of conservation with 1.00 corresponding to absolute conservation. (**f**) and (**g**): the electrostatic properties of the gp61.3-gp5Lys interface (an open book representation). (**h**) The electrostatic properties of gp5_G322D_ mutant. The electrostatic potential was calculated using ABPS [49].

**Figure 10 viruses-12-01070-f010:**
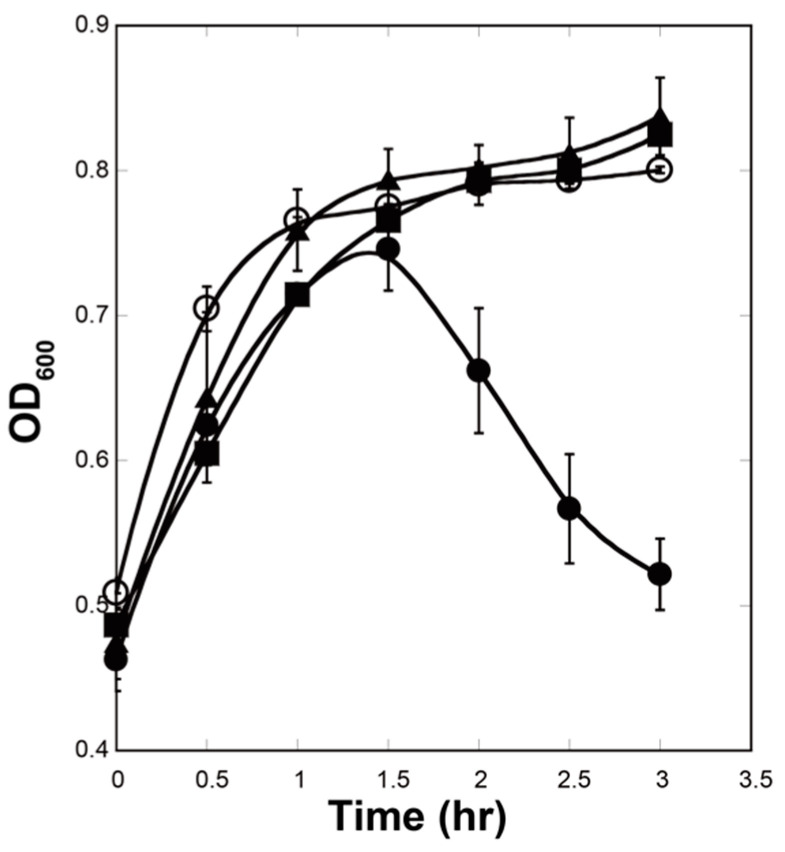
Full-length gp5 is translocated into the periplasmic space during overexpression. The cell turbidity assay of *E. coli* expressing gp5 (filled circles), T4L (filled triangle) and carrying a pET29a empty vector (filled squares). The cells were induced at *T* = 0 with 1 mM IPTG. Purified gp5 was added to *E. coli* cell culture harboring a pET29a vector (empty circles) at T = 0. The cell turbidity was measured at a wavelength of 600 nm. The symbols correspond to the averages and the bars to the standard deviations of *n* = 3 experiments.

**Figure 11 viruses-12-01070-f011:**
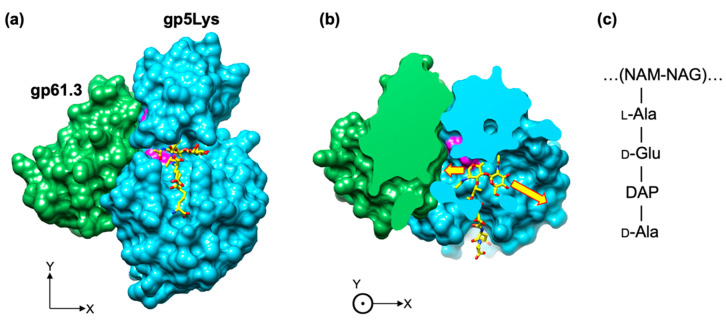
Gp61.3 interferes with free access of the polysaccharide moiety of the peptidoglycan substrate to the gp5Lys active site cleft. (**a**) and (**b**): two orthogonal views of the gp61.3-gp5Lys complex in the molecular surface representation. One repeating unit of the peptidoglycan is shown as a ball-and-stick model with C atoms in yellow, N in blue, and O in red. The active site residues of gp5 (Glu184, Aps193, Thr199) are colored magenta. In panel (**b**), the front part of the model is removed (slabbed away) to reveal the substrate-binding cleft. The yellow arrows indicate the directions in which the polysaccharide moiety of the peptidoglycan extends away from the presented model. (**c**) The chemical structure of the repeating unit of the peptidoglycan, the substrate of gp5. The continuation of the sugar moiety is shown with ellipses. NAM, NAG and DAP stand for N-acetylmuramic acid, N-acetylglucosamine and meso-diaminopimelic acid, respectively.

**Table 1 viruses-12-01070-t001:** X-ray data collection, reduction, and refinement statistics for gp61.3 and gp61.3-gp5Lys complex structures.

Crystal	Gp61.3 SeMet	Gp61.3-gp5Lys
**Data collection**		
Wavelength (Å)	0.97941	1.0000
Number of frames	325	360
Frame width (°)	1.0	1.0
Space group	P2_1_	P2_1_2_1_2_1_
Cell dimensions a, b, c (Å)	30.376, 46.921, 75.277	46.72, 69.18, 83.82
Resolution (Å)	39.8–1.60 (1.64–1.60) *	17.92–1.15 (1.21–1.15)
R_meas_	0.064 (0.158)	0.109 (1.885)
<I>/<σ_I_>	16.0 (5.8)	12.2 (1.8)
Completeness (%)	94.2 (73.7)	99.7 (99.8)
Multiplicity	3.3 (3.1)	13.5 (12.6)
Anomalous signal ^#^	4.91 (2.42)	N.A. ^$^
**Refinement**		
Resolution	19.90–1.60	17.92–1.15
No. reflections used in refinement	51,366	97,671
No. atoms (non-H)	2272	2630
Protein	1870	2195
Ligand/ion	3	53
Water	399	382
R_work_/R_free_	0.179/0.228	0.107/0.122
Average B factor (Å^2^)	18.0	16.3
Protein	15.6	13.5
Ligand/Ion	33.1	32.7
Water	29.1	30.1
R.m.s. deviations		
Bond lengths (Å)	0.002	0.008
Bond angles (°)	0.493	1.066
Ramachandran plot statistics		
Favored (%)	99.01	97.39
Allowed (%)	0.91	1.61
Outliers (%)	0.00	0.00
PDB accession code	7CN6	7CN7

* Statistics for the highest resolution shell is shown in the parentheses. ^#^ As defined by the program SHELXD [16]. ^$^ The structure was solved by molecular replacement.

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
