# Peer review of "Structure and Function of the T4 Spackle Protein Gp61.3"

_viruses, 2020, doi:10.3390/v12101070_

Round 1

Reviewer 1 Report

Summary:

Kanamaru et al. report the novel structure and function of an elusive T4 phage protein gp61.3 (Spackle). First identified over 50 years ago, structural characterization of gp61.3 appears to be the missing piece needed to answer multiple outstanding questions within the T4 field surrounding the function of a lysozyme domain within the gp5 tail protein complex. They demonstrate that gp61.3 is localized to the periplasm when expressed in E. coli and analyze the interaction between the gp5 lysozyme domain and gp61.3 using analytical ultracentrifugation. They also show that gp61.3 specifically inhibits gp5Lys and not the other T4 lysozyme, T4L, using a lysozyme halo formation assay. They report a crystal structure of gp61.3 for the first time, as well as a co-crystal structure of gp61.3 and gp5Lys. These structures demonstrate that gp61.3 adopts a novel fold, and possibly employs a novel inhibition mechanism: sterically blocking substrate association to the active site, and locking gp5Lys into a static, catalytically inactive conformation. The co-crystal structure also reveals that a revertant mutation in gp5 first identified in 1980 (T4-5ts1) is caused by the disruption of this interaction. Overall, I found the paper to be interesting, and it appears to be historically important to the T4 field, as it definitively answers some long-standing questions. The crystallographic work that makes up a bulk of the paper is of high quality.

Minor points:

Figure 3: If possible, it would be nice to see AUC data for gp5Lys, gp5G322D1-372, and T4L as well since these proteins appear later on and factor into the explanation of mechanism.   

Line 356: Please state the RMSD values for the Gly96 and Glu97 deviations between the two gp61.3 structures.

Figure 7: Figure is small and hard to see. In (a), the ball and stick models of interacting side chains are difficult to see or interpret as shown. Figure 9 shows the region of interaction more effectively. Perhaps consider showing the cartoon model and include a 180 degree rotated view. Referring to the “jaws” as N- and C-terminal is a little confusing because on line 367 it is stated that the lobes are made of residues: 166-184 and 248-342; and 185-247, respectively. Perhaps consider renaming these jaw A and B or something along these lines.

Figure 7 (continued): For (b) and (c), the inclusion of the peptidoglycan unit is a little confusing because this is one of the first figures showing the gp61.3:gp5Lys co-crystal structure. By presenting the docked PG substrate this early, and without explicit explanation in the figure caption, a reader could assume that the substrate was part of the co-crystal structure until this is clarified in line 388, a page after the structures are initially introduced. 

Figure 10: There is no corresponding methods section for this assay. Why is the starting OD of open circles (gp5 in culture media) so much higher than the rest?

Line 513: Since the substrate binding pocket is not biochemically analyzed in significant detail in this study, it is difficult to judge how convincing the argument for a novel mechanism is. If I am interpreting of the interaction between gp5Lys and peptidoglycan correctly, the substrate binding pocket would be across the entire central groove of the protein, best seen in Figure 9. This means that gp61.3 is acting as a plug on one end of the groove, while also holding the “jaws” open. The inhibition mechanism would therefore also be driven by steric clash between the sugar polymer and gp61.3, which may prevent entry of peptidoglycan into the groove containing the active site. Therefore, I would hesitate to describe this as an allosteric inhibition mechanism without additional data, for instance binding kinetics, and quantitative measurements of gp5Lys enzymatic properties, etc. I would recommend toning down the discussion/conclusions to reflect this.

Figure 1: It is unusual to present a figure in the introduction. Is it possible to shift this to the first section of results?

Lines 267-277: The rationale for the tryptic digest experiment should be stated. Was this done to obtain a more stable fragment amenable to crystallography?

Figure 5: Some of the labelled residues are never mentioned in the paper, so highlighting them may be confusing to readers.

Lines 394-404: Consider moving the proposed inhibition mechanism section to the discussion because you don’t have direct evidence for a lot of these claims, and you are making inferences from the literature.

Lines 405-410: Not clear from these global averages what the differences are, especially since only the active site of T4L and gp5 are conserved. It might help to include a supplementary figure of these structural overlays.

Line 507: The overlap of Gp5C and gp61.3 binding sites should be brought up in the results when they are shown together. This is an interesting observation that should be highlighted.

Author Response

Response to Reviewer 1 Comments

Kanamaru et al. report the novel structure and function of an elusive T4 phage protein gp61.3 (Spackle). First identified over 50 years ago, structural characterization of gp61.3 appears to be the missing piece needed to answer multiple outstanding questions within the T4 field surrounding the function of a lysozyme domain within the gp5 tail protein complex. They demonstrate that gp61.3 is localized to the periplasm when expressed in E. coli and analyze the interaction between the gp5 lysozyme domain and gp61.3 using analytical ultracentrifugation. They also show that gp61.3 specifically inhibits gp5Lys and not the other T4 lysozyme, T4L, using a lysozyme halo formation assay. They report a crystal structure of gp61.3 for the first time, as well as a co-crystal structure of gp61.3 and gp5Lys. These structures demonstrate that gp61.3 adopts a novel fold, and possibly employs a novel inhibition mechanism: sterically blocking substrate association to the active site, and locking gp5Lys into a static, catalytically inactive conformation. The co-crystal structure also reveals that a revertant mutation in gp5 first identified in 1980 (T4-5ts1) is caused by the disruption of this interaction. Overall, I found the paper to be interesting, and it appears to be historically important to the T4 field, as it definitively answers some long-standing questions. The crystallographic work that makes up a bulk of the paper is of high quality.

Thank you very much for such a nice summary.

Point 1: Figure 3: If possible, it would be nice to see AUC data for gp5Lys, gp5G322D1-372, and T4L as well since these proteins appear later on and factor into the explanation of mechanism.   

Response 1: We analyzed the complex formation (the lack of thereof) of T4L and gp61.3 using size exclusion chromatography. See Supplementary Figure S2.

Point 2: Line 356: Please state the RMSD values for the Gly96 and Glu97 deviations between the two gp61.3 structures.

Response 2: Done.

Point 3: Figure 7: Figure is small and hard to see. In (a), the ball and stick models of interacting side chains are difficult to see or interpret as shown. Figure 9 shows the region of interaction more effectively. Perhaps consider showing the cartoon model and include a 180 degree rotated view. Referring to the “jaws” as N- and C-terminal is a little confusing because on line 367 it is stated that the lobes are made of residues: 166-184 and 248-342; and 185-247, respectively. Perhaps consider renaming these jaw A and B or something along these lines.

Response 3: We re-rendered the panel (a) as it was too foggy and rearranged the panels so that each panel is larger. The jaws of gp5Lys has been renamed.

Point 4: Figure 7 (continued): For (b) and (c), the inclusion of the peptidoglycan unit is a little confusing because this is one of the first figures showing the gp61.3:gp5Lys co-crystal structure. By presenting the docked PG substrate this early, and without explicit explanation in the figure caption, a reader could assume that the substrate was part of the co-crystal structure until this is clarified in line 388, a page after the structures are initially introduced. 

Response 4: We moved the description of the inhibition mechanism and the relevant panels to Discussion.

Point 5: Figure 10: There is no corresponding methods section for this assay. Why is the starting OD of open circles (gp5 in culture media) so much higher than the rest?

Response 5: The missing Methods section has been added. The higher cell density was an oversight. The experiment was redone with the appropriate cell density.

Point 6: Line 513: Since the substrate binding pocket is not biochemically analyzed in significant detail in this study, it is difficult to judge how convincing the argument for a novel mechanism is. If I am interpreting of the interaction between gp5Lys and peptidoglycan correctly, the substrate binding pocket would be across the entire central groove of the protein, best seen in Figure 9. This means that gp61.3 is acting as a plug on one end of the groove, while also holding the “jaws” open. The inhibition mechanism would therefore also be driven by steric clash between the sugar polymer and gp61.3, which may prevent entry of peptidoglycan into the groove containing the active site. Therefore, I would hesitate to describe this as an allosteric inhibition mechanism without additional data, for instance binding kinetics, and quantitative measurements of gp5Lys enzymatic properties, etc. I would recommend toning down the discussion/conclusions to reflect this.

Response 6: Unlike any previously described lysozyme inhibitor, gp61.3 does not interact with the substrate cavity directly. That is – all known inhibitors insert an ‘arm’-like structure into the active site cleft (see e.g. Ref. 59). Gp61.3 does not do that. Instead, it creates a wall at one end of the active site cleft. T4L and, by homology, gp5Lys are endoglycosidases. With one side of the sugar-binding cleft obstructed, the enzymes might turn into exoglycosidases. Gp61.3 inhibits the activity of gp5Lys in a rather dramatic fashion, so we believe it is not an endo-to-exo conversion. Gp61.3 puts a break on gp5Lys activity without blocking the active site. Our analysis suggests that it does so by keeping the gp5Lys jaw domains wide open, which is an allosteric mechanism by definition.

Point 7: Figure 1: It is unusual to present a figure in the introduction. Is it possible to shift this to the first section of results?

Response 7: This figure summarizes previous findings that are described in detail in Ref. 12 and 13 and introduces new gp5 constructs that are utilized in this study. We have changed the figure and the corresponding figure legend to make the figure and its placement in the Intro easier to understand.

Point 8: Lines 267-277: The rationale for the tryptic digest experiment should be stated. Was this done to obtain a more stable fragment amenable to crystallography?

Response 8: Thanks for pointing this out. This has been corrected.

Point 9: Figure 5: Some of the labelled residues are never mentioned in the paper, so highlighting them may be confusing to readers.

Response 9: The figure legend states: The start and end residues for each helix are indicated. We believe it is easier to trace the chain in 3D with those labels shown.

Point 10: Lines 394-404: Consider moving the proposed inhibition mechanism section to the discussion because you don’t have direct evidence for a lot of these claims, and you are making inferences from the literature.

Response 10: Done.

Point 11: Lines 405-410: Not clear from these global averages what the differences are, especially since only the active site of T4L and gp5 are conserved. It might help to include a supplementary figure of these structural overlays.

Response 11: In our 2002 Nature paper (ref 13), the comparison of gp5Lys with T4L is discussed in greater detail. The overall structures of T4L and gp5Lys, not just the active sites, are nearly identical as shown by the very small RMSDs found on superposition. Granted, this might be confusing as we failed to mention the percentage of atoms that participate in the alignment (it is 96%: 159 equivalent Calpha atoms out of 164 residues comprising T4L and 168 residues comprising gp5Lys). This value is given in the new version of the text.

We see little value in showing an overlay of two nearly identical structures such as T4L and gp5Lys (especially considering that this has been already discussed 18 years ago). To the contrary, as the structures are so similar, only the total superposition RMSD can capture the small differences associated with conformational changes.

Point 12: Line 507: The overlap of Gp5C and gp61.3 binding sites should be brought up in the results when they are shown together. This is an interesting observation that should be highlighted.

Response 12: We prefer to leave these sentences in the (new version of the) Discussion alongside the discussion of the functional mechanism.

Reviewer 2 Report

The manuscript by Kanamaru et al. uses structural analysis augmented by few biochemical and biophysical experiments to show that T4 Spackle protein interacts with the tail associated gp5 lysozyme and unraveled a novel, two-pronged inhibitory mechanism. They solved the structure of the periplasmic C-terminal (minus signal peptide) portion of Spackle and its complex with the tail N-terminal lysozyme fragment to high resolution. Comparison of the complex structure with those of other lysozymes revealed that Spackle inhibits gp5 by a dual action, first by obstructing substrate entry and second by preventing the substrate banding site closure. The structural analyses, apart from a failed attempt at thermodynamic interpretation (see specific comments below), are sound and well presented. However, the presentation and interpretation of the supporting biochemical and biophysical experiments lacks rigor and needs some improvement prior to publication, see specific comments and suggestions below. Overall, the presented results are novel and of interest not only to phage virologists but also to protein scientists since the related T4 lysozyme constitutes one of the best model for protein folding and stability and the present work describes a new  allosteric regulatory mechanism for this ubiquitous class of enzymes.

Specific issues:

  • Lines 241-242: Add reference to freeze-thaw procedure to release periplasmic proteins. Did you perform or obtained a positive control, e.g. identified a common periplasmic protein, such as SurA, being released to the media?
  • There is certainly something wrong with he labelling of bands in Fig.2: gp61.3 in lane 2 and 3 have quite different migration, why? Also , how do you know it is the right protein, esp. in lanes 1-2, where there might be many cellular proteins migrating at the same spot. It is customary to show cell lysate prior to IPTG induction (negative control). What about a simple Western?
  • Lanes 263-266: Sedimentation coefficient does depend on both mass and shape so this analysis uses some assumptions about the shape. However, having high res structures permits to compute S values (and other parameters) directly from PDBs of the components and the complex and thus link the solution sedimentation analyses (which is robust and convincing) rigorously with the structures. For example program HYDRO does this but there might be web based apps. This will also address any lingering issues about stoichiometry.
  • Line 265-266: No data shown for the control, please add it to the supplement.
  • Lines 273-74: Was mass spec done to confirm that the C-terminus remained intact upon trypsin digest?
  • Figure 4: IN the intro the authors decide to use T4L instead of gp e (lanes 35-36) so it would be better to stick with this in the figure labelling as well.
  • The (pseudo) thermodynamic interpretation of the structural information (lanes 348-353) is confusing, and possibly wrong due to the choice of different reference state for computation of association and dissociation energies (BTW there is no information about how this computation was done in the Methods). Hence the results do not square with the convincing sedimentation data which show stable complex formation at 1:1 stoichiometry. I suggest to limit comparison to the buried areas and perhaps augment this section by comparing the interfacial area with those found in very stable and moderately stable inhibitory complexes (e.g. serpin-protease complexes).
  • Line 358: Given that calcium seems to bind to the monomeric Spackle I would expect it to inhibit complex formation. Please, comment and reconcile.
  • Lines 354-55: replace “slightly but not insignificantly” with “slightly but significantly”
  • Figure 7 legend lines 382-83: There are no ellipses in the figure, perhaps wrong version of the legend?
  • Figure 9: It seems that most of the residues at the interface are conserved so what is so special about G322 if it is equally conserved? Perhaps it is its location in the middle of the interface so any bulky substitution would be hard to accommodate without remodeling large portion of the adhering surfaces on both proteins. Since the mutation also introduces neg. charge it would be nice to see the electrostatic potential of the complementary interface instead of the rather meaningless conservation map. This analysis can certainly be improved.

Author Response

Response to Reviewer 2 Comments

Comments and Suggestions for Authors

The manuscript by Kanamaru et al. uses structural analysis augmented by few biochemical and biophysical experiments to show that T4 Spackle protein interacts with the tail associated gp5 lysozyme and unraveled a novel, two-pronged inhibitory mechanism. They solved the structure of the periplasmic C-terminal (minus signal peptide) portion of Spackle and its complex with the tail N-terminal lysozyme fragment to high resolution. Comparison of the complex structure with those of other lysozymes revealed that Spackle inhibits gp5 by a dual action, first by obstructing substrate entry and second by preventing the substrate banding site closure. The structural analyses, apart from a failed attempt at thermodynamic interpretation (see specific comments below), are sound and well presented. However, the presentation and interpretation of the supporting biochemical and biophysical experiments lacks rigor and needs some improvement prior to publication, see specific comments and suggestions below. Overall, the presented results are novel and of interest not only to phage virologists but also to protein scientists since the related T4 lysozyme constitutes one of the best model for protein folding and stability and the present work describes a new  allosteric regulatory mechanism for this ubiquitous class of enzymes.

Thank you very much for your careful reading and thoughts on how to improve the MS.

Point 1: Lines 241-242: Add reference to freeze-thaw procedure to release periplasmic proteins. Did you perform or obtained a positive control, e.g. identified a common periplasmic protein, such as SurA, being released to the media?

Response 1: The relevant references [39, 40] have been added. We have not done the suggested control experiment. However, there is a tremendous enrichment of gp61.3 in the supernatant fraction after just a single round of freezing and thawing. Please compare the ratio of gp61.3 to other proteins (including the small ones) in the total cell lysate (Fig. 2, Lane 1 and Supplementary Fig. S1) to that after just one freeze-and-thaw cycle (Fig. 2, Lane 2). The simplest explanation is that essentially all gp61.3 that has been expressed in the cell, is located in the periplasm and it comes out easily through the holes in the outer membranes formed upon freezing and thawing.

Point 2:There is certainly something wrong with he labelling of bands in Fig.2: gp61.3 in lane 2 and 3 have quite different migration, why? Also , how do you know it is the right protein, esp. in lanes 1-2, where there might be many cellular proteins migrating at the same spot. It is customary to show cell lysate prior to IPTG induction (negative control). What about a simple Western?

Response 2: Fig. 2 shows several gels of different percentages of acrylamide that were run for different amounts of time. Hence, the proteins in question are located in different regions of the image. We enhanced the contrast of molecular weight markers to help with the interpretation. The pre- and post-induction samples are shown in Supplementary Fig. S1.

We do not have anti-gp61.3 antibodies and making new ones in 10 days allotted for revision is impossible. So, we were unable to do a WB. However, we do not understand the request for a WB as we present here a truly atomic resolution structure of gp61.3 (Table 1). In an electron density of this quality and at this resolution, not only side chains as a whole, but the identity of each atom can be established (see Fig. 6). Certainly, a protein can be sequenced de novo using the electron density alone.

Point 3: Lanes 263-266: Sedimentation coefficient does depend on both mass and shape so this analysis uses some assumptions about the shape. However, having high res structures permits to compute S values (and other parameters) directly from PDBs of the components and the complex and thus link the solution sedimentation analyses (which is robust and convincing) rigorously with the structures. For example program HYDRO does this but there might be web based apps. This will also address any lingering issues about stoichiometry.

Response 3: We do not think that there are any “lingering issues about the stoichiometry”. The AUC data are in perfect agreement with the atomic resolution crystal structure. We nevertheless ran HYDROPRO on the crystal structure and obtained a sedimentation coefficient of 3.66S (the AUC gave us 3.60, Fig. 3). This is mentioned in the new version of the text immediately after the initial description of the crystal structure. 

Point 4: Line 265-266: No data shown for the control, please add it to the supplement.

Response 4: We did not do AUC for the T4L-gp61.3 mixture. But we did do size exclusion chromatography. See Supplementary Fig. S2.

Point 5: Lines 273-74: Was mass spec done to confirm that the C-terminus remained intact upon trypsin digest?

Response 5: Again, no, we did not do that. The relevant sentences have been revised.

Point 6: Figure 4: IN the intro the authors decide to use T4L instead of gp e (lanes 35-36) so it would be better to stick with this in the figure labelling as well.

Response 6: Indeed. Thanks for catching this.

Point 7: The (pseudo) thermodynamic interpretation of the structural information (lanes 348-353) is confusing, and possibly wrong due to the choice of different reference state for computation of association and dissociation energies (BTW there is no information about how this computation was done in the Methods). Hence the results do not square with the convincing sedimentation data which show stable complex formation at 1:1 stoichiometry. I suggest to limit comparison to the buried areas and perhaps augment this section by comparing the interfacial area with those found in very stable and moderately stable inhibitory complexes (e.g. serpin-protease complexes).

Response 7: The function of the program PISA used here (originally) for the computation of complexation energy is described in Ref. [45]. PISA has been trained on a large number of interfaces (although it does not employ a neural network algorithm) and is fairly reliable at predicting ‘solution-worthy’ interfaces. Not so in this case, and we simply wanted to point this out.

Nevertheless, these comments and remarks compelled us to investigate the energetics of gp61.3-gp5Lys complexation in greater detail by using state-of-the-art computational tools (the Texas Advanced Computing Center, one of the most powerful non-military supercomputers in the world) and approaches (steered molecular dynamics and adaptive biasing force method). Please see Fig. 7 (b) and section 2.13 in the Methods. We found that the complexation energy is fairly substantial at about -14 kcal/mol.

Point 8: Line 358: Given that calcium seems to bind to the monomeric Spackle I would expect it to inhibit complex formation. Please, comment and reconcile.

Response 8: Calcium does not appear to play a role in complex formation. See Supplementary Fig. S3.

Point 9: Lines 354-55: replace “slightly but not insignificantly” with “slightly but significantly”

Response 9: Done.

Point 10: Figure 7 legend lines 382-83: There are no ellipses in the figure, perhaps wrong version of the legend?

Response 10: Here, the word ‘ellipses’ is the plural form of the word ‘ellipsis’.

Point 11: Figure 9: It seems that most of the residues at the interface are conserved so what is so special about G322 if it is equally conserved? Perhaps it is its location in the middle of the interface so any bulky substitution would be hard to accommodate without remodeling large portion of the adhering surfaces on both proteins. Since the mutation also introduces neg. charge it would be nice to see the electrostatic potential of the complementary interface instead of the rather meaningless conservation map. This analysis can certainly be improved.

Response 11: The electrostatic potential properties of the gp61.3-gp5Lys interface are shown in Fig. 9 (f) and 9 (g). The influence of the G322D mutation on the distribution of potential on the interface surface is shown in Fig. 9 (h). D322 neutralizes a strong positive patch in that location.

We would very much prefer to keep the conservation map panels because they are needed to support our supposition that gp61.3 inhibits gp5 orthologs of T4 relatives (relatively close relatives).

Round 2

Reviewer 2 Report

The authors did a thorough job addressing all my queries and concerns. The addition of steered MD and ABF greatly enhanced structural interpretation of the interactions. Well done, thank you.